# Early life exposures associated with risk of small intestinal neuroendocrine tumors

James VanDerslice[1], Marissa C. Taddie[1], Karen Curtin[2,3], Caroline Miller[1], Zhe Yu[3], Rachael Hemmert[1], Lisa A. Cannon-Albright[2,3,4], Deborah W. Neklason[2,3]*

1 Division of Public Health, Department of Family and Preventative Medicine, University of Utah, Salt Lake City, Utah, United States of America, 2 Division of Epidemiology, Department of Internal Medicine, University of Utah, Salt Lake City, Utah, United States of America, 3 Huntsman Cancer Institute, University of Utah, Salt Lake City, Utah, United States of America, 4 George E. Wahlen Department of Veterans Affairs Medical Center, Salt Lake City, Utah, United States of America

* deb.neklason@hci.utah.edu

**Data Availability Statement:** We have indicated that data from this study are available upon request due to legal restrictions on sharing data publicly.

## Abstract

Small intestinal neuroendocrine tumors (SINT) are rare with incidence increasing over the past 40 years. The purpose of this work is to examine the role of environmental exposures in the rise of SINT incidence using the Utah Population Database, a resource of linked records including life events, cancer diagnoses and residential histories. SINT cases born in Utah were identified through the Utah Cancer Registry with: diagnosis years of 1948 to 2014 and age at diagnosis of 23 to 88 years. Controls were matched to cases 10:1 based on sex, birth year and residence time in Utah. Cases and controls were geocoded to their birth locale. An isotonic spatial scan statistic was used to test for the occurrence and location(s) of SINT clusters. Potential environmental exposures and economic conditions in the birth locales at the time of the birth (1883–1982) were generated using historical references. Conditional logistic regression was used to estimate odd ratios. We report a spatial cluster central to historic coal mining communities, associated with a 2.86 relative risk ($p = 0.016$) of SINT. Aspatial analyses of industry and mining exposures further suggest elevated risk for early life exposure near areas involved in the construction industry (OR 1.98 $p = 0.024$). Other exposures approached significance including coal, uranium and hard rock mining during the earliest period (1883–1929) when safety from exposures was not considered. We do observe a lower risk (OR 0.58 $p = 0.033$) associated with individuals born in rural areas in the most recent period (1945–1982). Environmental exposures early in life, especially those from industries such as mining, may confer an elevated risk of SINT.

## Introduction

Carcinoids are neuroendocrine tumors with the majority occurring in the gastrointestinal tract (67%). Incidence of carcinoid tumors is about 2.47 (0.87 for small intestine alone) per 100,000 population per year with the incidence having increased 3-fold over the past 40 years [1, 2]. Multiple factors likely contribute to the rise in incidence of small intestinal neuroendocrine (carcinoid) tumors (SINT) including better and more accurate detection, tobacco use,

Use of data from Utah Population Database are governed by Utah Resource for Genetic and Epidemiologic Research (RGE). The RGE was first established by Executive Order of the Governor of the State of Utah on July 14, 1982, and modified by a second Executive Order on February 20, 1986, as "a data resource for the collection, storage, study, and dissemination of medical and related information" for "the purpose of reducing morbidity or mortality, or for the purpose of evaluating and improving the quality of hospital and medical care." (Utah Code 26-25-1, 63G-2-3). The University of Utah administers RGE to facilitate appropriate access and responsible use of data by research and public health projects. RGE's primary role is the protection of the data contributor and data subject against inappropriate access and use of the data governed by RGE. Applications for UPDB data use can be sent to: Utah Resource for Genetic & Epidemiologic Research (RGE) Jahn Barlow, MPA, Director Research Administration Building 75 South 2000 East, Room 105 Salt Lake City, UT 84112 801-581-6351 rge@hsc.utah.edu https://rge.utah.edu/index.php.

**Funding:** This work was supported by grants from the National Cancer Institute at the National Institutes of Health, R21CA205796 and P30CA042014 and Huntsman Cancer Foundation. Support for the Utah Cancer Registry is provided by the National Cancer Institute's SEER Program, Contract No. HHSN261201800016I, the US Center for Disease Control and Prevention's National Program of Cancer Registries, Cooperative Agreement No. NU58DP0063200, with additional support from the University of Utah and Huntsman Cancer Foundation. Research was supported by the NCRR grant, "Sharing Statewide Health Data for Genetic Research" (R01 RR021746) with additional support from the Utah State Department of Health and the University of Utah.

**Competing interests:** The authors have declared that no competing interests exist.

and changes in environmental exposures [3–6]. Tobacco use has declined since its peak in the 1960's, however the lag time from tobacco exposure to cancer later in life is consistent with the recent rise in incidence of SINT [7]. Although SINT are slow growing tumors, they are often not diagnosed until advanced stage when prognosis is dramatically diminished. Advances in imaging have allowed for earlier detection, however, the asymptomatic and rare nature of this tumor makes broad screening impractical. The key to reversing the trend is to better understand risk factors, mitigate these risk factors, identify individuals at risk, and implement early diagnosis tools.

Because this is a rare cancer, epidemiologic studies have been limited. Carcinoid tumors comprise approximately 40% of all small intestinal primary tumors and are similar in frequency to adenocarcinomas of the small intestine [8–10]. Studies, including our recent publication, have consistently shown a high relative risk of 6- to 11-fold in first-degree relatives, with siblings having twice the relative risk of parents [9, 11, 12]. The increase in concordance between siblings versus parents suggests that some aspect of a shared environment compounds the risk of developing SINT. Cigarette smoking and alcohol have been associated with an elevated risk of SINT, but reports are varied. In a meta-analysis including patients from the U.S.A. and Europe, an odds ratio of 1.40 (1.06–1.86) was observed for smoking and 1.04 (0.63–1.72) for alcohol [6]. In Utah, we observed an odds ratio of 1.44 (1.11–1.86) for smoking and 1.62 (1.05–2.49) for alcohol and risk of SINT [13]. Utah has one of the lowest smoking and alcohol consumption rates in the United States due to the majority of the population being members of the Church of Jesus Christ of Latter-Day Saints (a.k.a. Mormon) which has proscriptions against drinking and smoking, however the rising incidence of SINT are comparable to national reports [1, 14], suggesting other risk factors are involved. Environmental exposure to carcinogens are considered to be a driving force in increased incidence of some cancers such as SINT [15].

A recent report using data from SEER suggests that the increasing incidence of colorectal cancer in younger adults is consistent with a "birth cohort effect and points to early life exposures, accumulated throughout the life course—that may increase cancer risk" [16]. Similar to the increasing incidence of colorectal cancer in younger adults, SEER data (1975–2016) shows a 3.4 annual percent increase in SINT incidence under age 50y (SEER*Stat 8.3.6). However, diagnoses of SINT over age 50y are also increasing (2.4 annual percent increase) while colorectal cancers in this age group have been decreasing. One of the more dramatic examples of early life exposure and elevated risk of solid cancers is a study of atomic bomb survivors who were exposed in utero or as young children (0–5 years) [17]. One of the challenges of studying the effects of early life conditions on subsequent adult cancer risks is the lack of location and environmental data during the relevant exposure period. For example, all SINT cases diagnosed in Utah were born before 1982 with 57% having been born before 1940. Data resources at the University of Utah offer a unique opportunity to explore the relationship between early life environmental exposures and subsequent cancer risks. Multiple studies have used these statewide resources to consider early life conditions including socioeconomic status, parental mortality, religion and birth order on long-term health outcomes including cancer, mortality and systemic inflammation [18–20]. Early life exposures based on place of birth has not, to our knowledge, been explored using the Utah state-wide resources.

Extensive data is available through the Utah Population Database (UPDB), a state-wide resource containing all individuals with a life event in Utah (e.g. birth, marriage, death, driver license renewal or hospitalization) with records linked to statewide cancer data [21, 22].

The objective of this study was to test for the occurrence and location(s) of SINT clusters based on birth location and assess the associations between early life environment exposures and SINT cases. We examined geographic clustering of SINT cases based on birth locale and exposure based on industries present at the time and place of birth.

## Materials and methods

Study was approved by the Institutional Review Board of University of Utah and the Utah Resource for Genetic and Epidemiologic Research Board (the governing body of UPDB use).

### UPDB data resource

UPDB is a shared resource located at the University of Utah and consists of computerized data records for >11 million deceased or living people dating from the 1700's to the present. (http://healthcare.utah.edu/huntsmancancerinstitute/research/updb/). Although the population is dynamic, more than 95% of current adult residents in Utah (2.3 million) are represented in UPDB. Individuals are included in the database if they have a recorded event which can be birth, marriage, death, Utah driver license renewal, inpatient and ambulatory facility administrative claims (statewide), or have a record in the Utah Cancer Registry (UCR). The UCR is a National Cancer Institute Surveillance Epidemiology and End Result (SEER) registry established in 1973 that has collected information on all cancers diagnosed in Utah from 1966 forward, and incomplete cancer data prior to 1966. These data are updated annually, and probabilistic record-linking is performed with individuals in UPDB as described [23]. The UPDB is the only database of its kind in the United States, and one of a few in the world; most families living in Utah are represented in the UPDB [24].

### Case and control selection

Cases were identified from the UCR with a SINT diagnosis (International Classification of Diseases-for Oncology, 3rd Edition codes C170-C173 or C178-C179 and a histology code of 8240 or 8241) from 1948 to 2014 (n = 745). Three hundred and ninety cases fulfilled the inclusion criteria of individuals born in Utah. Eight were diagnosed before the establishment of the UCR in 1966 (2%). There was no geographical clustering of these 8 early cases; they represented 6 counties, both urban and rural, and were similar in distribution of age and sex to the cases diagnosed after 1966. Controls were randomly selected from among individuals in the UPDB who were born in Utah and matched 10:1 to cases based on sex, birth year, and residence time in Utah in years based on their earliest and latest record in UPDB (n = 3710). Race and ethnicity were derived from birth certificates or other UPDB data sources. We assessed tobacco and alcohol exposure from International Classification of Diseases 9th or 10th Revision (ICD-9, ICD-10) codes appearing in any diagnostic position in one or more statewide Utah Department of Health and University of Utah Health or Intermountain Health electronic medical records from 1996 forward and self-reported tobacco and alcohol histories taken routinely at each inpatient or outpatient encounter in a University of Utah Health or Intermountain Health facilities from 2009 as described previously [25]. For cases and controls who died prior to 1996, contributing cause of death from tobacco or alcohol noted on a Utah death certificate was used to provide an indication of use. Cases and controls with any positive indication of tobacco or alcohol use in the medical record (1 or more encounters) or from a Utah death certificate were assigned "Yes indicated" and otherwise assigned "Not indicated". The majority of cases and their corresponding controls (86%) had medical records in UPDB for 1996 or later. A total of 4100 cases and controls, assigned to 405 birth locales, were included in the geographic analysis.

### Geocoding

Cases and controls from Utah were born between 1883 and 1982 and were geocoded to their birth locale. Ascertaining birth locale was based on the mother's residential address reported

on the birth certificate (95% of cases and 97% of controls), and when that was unavailable, we used the "birth city" on the birth certificate (usually the hospital location; 4.29% of cases and controls) or a birth location derived by the UPDB using other linked records from the same time period. Salt Lake City is the largest city in Utah and was the birth place for nearly 25% of cases and controls. In order to make Salt Lake City more comparable in population and area to the other birth cities, Salt Lake City's historic neighborhood boundaries were delineated using information from Salt Lake City's master plan areas, community council areas, local historic districts, and national historic districts [26, 27]. This resulted in 39 neighborhoods that encompassed the municipal boundaries of Salt Lake City. A U.S. Address-City State locator was created for the project and used to geocode birth locales and Salt Lake City street addresses using ArcGIS Desktop Release 10.3.1 (Environmental Systems Research Institute, 2016, Redlands, CA, USA). The address locator developed for the project not only included contemporary cities, but also historic mining, agricultural and railroad towns, as well as historic neighborhoods and religious congregation districts. This collection of diverse settlements associated with residence at birth we refer to as a birth locale. In order to represent the geography of birth locales as they occurred through time, historic towns that are now part of a single city were not aggregated. In addition, it was necessary to standardize birth locale names since many settlements had earlier name variations. Utah Place Names [28], Utah Division of State History [29], and Geographic Names Information System [30] were the primary sources used to assign geographic coordinates to birth locales. Cases and controls with addresses from Salt Lake City were geocoded to the centroid of their respective neighborhood and those that could not be assigned to a neighborhood (54 cases; 470 controls) were aggregated to the Salt Lake City centroid. A total of 4100 cases and controls were geocoded to 405 birth locales.

## Geographic analysis

An isotonic spatial scan statistic (ISSS) was implemented in SaTScan (v9.6) to test for the occurrence and location(s) of SINT clusters using the Bernoulli probability model [31–33]. Cases and controls were aggregated to their birth locales. Since birth year was one of the matching variables for cases and controls, it was not adjusted for in the ISSS model. The ISSS scans the region for clusters using a collection of variably sized circles centered on each birth locale. There is no limit on the number of circles, only that the largest circle is smaller than the user-defined maximum circle size, which in our case, was up to 50% of the population at risk. The alternative hypothesis when using ISSS is that the estimated risk is greatest in the innermost circle and decreases continuously to the outermost circle with no a priori assumptions about the number of decreasing steps [32, 33]. The collection of circles with the maximum likelihood ratio is reported as the most likely cluster. A significant isotonic cluster has an overall relative risk (RR), which, for this study, is the risk in the overall cluster compared to the rest of Utah, as well as a RR for each cluster step, which is the risk of each cluster step compared with the rest of Utah. To evaluate the occurrence of smaller clusters, while still adjusting for the presence of larger clusters, the maximum reportable cluster size was incrementally increased from 10% to 40% of the population at risk in ten-percent intervals. Statistical significance was assessed using Monte Carlo hypothesis testing [34].

## Exposure assignment based on birth locale history

The relationship between early life exposures and the risk of SINT was examined using a case-control design as described above. Time-varying historical profiles, covering the time span of births (1883 to 1982) were constructed for each of the 405 birth locales. These profiles included information on potential exposures such as the major manufacturing and industrial

operations, mining and smelting operations, and transportation of coal and ore. Agricultural activities (i.e. orchards, truck crops, dry farming and livestock) and settlement size and connectedness to major metropolitan areas were also determined for each birth locale throughout the range of birth years. City was designed to capture whether the subject was born in a significant economic or political settlement of the area for the time period of the birth. Rural was designed to capture areas that were not in proximity to the major metropolitan centers. Locales that are neither city nor rural are small towns that while not contiguous with a metropolitan area, are also not isolated from an urban area. This information was abstracted from multiple historical documents including the Utah Centennial County History Series, a set of comprehensive histories for each county, mining and railroad histories, and US Environmental Protection Agency documentation [28, 29, 35–39]. For each of these activities we used the data in the historical documents to estimate the year that the activity began and the year it ceased. In most cases, the birth locales were relatively small and documents about the history of the area clearly described the major industries and economic activities along with a general description of the years these activities began and ended. We also created a comprehensive georeferenced database of all major mines and smelters and used this to identify birth locales within these industries.

Birth locale profiles included 16 dichotomous exposure indicator variables denoting the presence of each potential exposure or characteristic with starting and ending year noted. The exposures and definitions are described in Table 1 and detailed information regarding the sources of historical information are presented in S1 Table. A birth locale could have more than one of these exposure indicators at any point in time, and the exposures attributed to a

**Table 1. Industries, agriculture and settlement type assigned to birth locale.**

| **Mining** | |
| --- | --- |
| Any Mining | Any type of mining regardless of commodity type. |
| Hard Rock | Mining of any metal, coal or uranium, or quarry processing of rocks such as limestone. |
| Coal | Mining, processing or transportation hub for coal. |
| Uranium | Mining or processing of uranium. |
| Minerals | Mining and processing of softer rocks and minerals, such as gypsum or ozorkerite. |
| **Industry** | |
| Heavy Industry | Any heavy machine-based or complex process manufacturing including ore processing, heavy construction, large industrial plants, or extractive practices. |
| Smelting | Any extraction of metal from ore by applying heat. |
| Construction | Any large scale construction of highways, dams, or public works. |
| **Agriculture** | |
| Crop | Any mention of dry or irrigation based agriculture, commonly hay, grains and sugar beets. |
| Orchard | Any mention of fruit bearing crops and/or fruit bearing trees. |
| Livestock | Indicates ranching, grazing, or animal feeding operations. |
| **Type of settlement** | |
| City | Locales that were the significant economic or political hub for an area during the year of birth. Includes locales contiguous with these areas. |
| Rural | Locales that were not contiguous with, or a suburb of one of the major metropolitan areas at that time period. |
| Commerce | Denoted by use for trade or commercial good exchange. |
| Connected | Areas established along railroad sidings and highway junctions, largely transportation based industries. |
| Shipping | Any location primarily used for distribution centers, especially railroad use. |

locale varied over time. The exposure indicators were assigned to cases and controls based on their birth locale and year.

Conditional logistic regression was used to estimate both crude odd ratios (OR) and adjusted odds ratios (aOR) with adjustment for indication of smoking and alcohol consumption and family history of SINT. The analysis was stratified based on three historic time periods, 1883 to 1929 (Period 1), 1930 to 1944 (Period 2) and 1945 to 1982 (Period 3) to account for changes over time in potential exposures associated with a specific industry. For example, worker or community exposures associated with coal mining may have changed due to improvements in occupational safety practices or extraction and processing improvements. The time periods were defined based on major economic transitions that affected many industrial processes, namely the Great Depression and the end of World War II. Two cases and seventeen controls were excluded from the adjusted analysis due to lack of family history, defined as having no second-degree relatives in UPDB. Given the small sample size and exploratory nature of this analysis we did not adjust for multiple comparisons.

## Results

A birth locale was determined for 390 cases and 3710 matched controls (Table 2). Race, indication of tobacco use, and indication of alcohol use have similar proportions between cases and controls. As expected, a greater proportion of cases had a family history of SINT. In general, the distribution of tumor location and stage are similar to our Utah data of all cases (n = 745), including those born outside of Utah as well as national SEER data. The mean and median age of diagnosis in this study was two years younger than reported for national SEER data, but identical to our overall Utah data [1]. Tumor grade was reported on only 29% of the cases, with the majority being well differentiated.

For our spatial analysis, 405 birth locales were used representing all cases and controls born in Utah from 1883 until 1982. Most of the locales (60.5%) were the birth place of a single SINT case. Three locales, Salt Lake City (neighborhood undetermined, n = 54), Ogden (n = 21), and Logan (n = 12) were the birth place for 22% (87/390) of the SINT cases.

One significant isotonic (spatial) cluster occurred in Utah. The isotonic cluster has three steps as well as an overall relative risk (RR) for the whole cluster as shown in Fig 1. For each step, the RR is the risk within the cluster step as compared to the area outside of the cluster, i.e., the rest of Utah. The first step consists of one birth locale containing four cases, and is at the center of the cluster with a RR of 8.58. The relative risk was attenuated in the second step of the isotonic cluster, which had an RR of 6.31 and included four locales and three additional cases. In Step 3, the RR decreased to 1.53 and contained eight locales and five additional cases. The overall RR of the isotonic cluster was 2.86 and had the same areal extent as Step 3 (Fig 1). Twelve SINT cases were born in this geographic area, whereas 4.28 cases would be expected ($p$ = 0.016). Interestingly, all of the cases were born prior to 1940 (1897–1935). As shown in Fig 1, this region was historically a coal mining and shipping region with little agriculture or manufacturing [35].

The association of environmental characteristics with SINT was investigated using a time-stratified case-control analysis of 16 exposure categories of the birth locale (Table 3). Those born in a city with a large construction project during Period 1 (1883–1929) had a significantly greater risk of SINT (OR = 1.98, 95% confidence interval (CI) 1.09 to 3.60, $p$ = 0.024). During Period 1 there was also a suggestive but not significant increases in odds of SINT cases being born in a locale with mining activities, including coal mining (1.95 with 95% CI of 0.97 to 3.92, $p$ = 0.061), hard rock mining (1.38 with 95% CI of 0.97 to 1.97, $p$ = 0.075), and uranium mining (2.70 with 95% CI of 0.86 to 8.48, $p$ = 0.088). An elevated, but not significant, odds ratio

**Table 2. Demographics of cases and controls.**

|  | Cases (390) |  | Controls (3710) |  |
|---|---|---|---|---|
| Gender |  |  |  |  |
| Male | 231 | 59% | 2215 | 60% |
| Female | 159 | 41% | 1495 | 40% |
| Birth Year |  |  |  |  |
| 1883–1929 (Period 1) | 147 | 38% | 1359 | 37% |
| 1930–1944 (Period 2) | 116 | 30% | 1123 | 30% |
| 1945–1982 (Period 3) | 127 | 33% | 1228 | 33% |
| Race |  |  |  |  |
| White or Caucasian | 389 | 100% | 3637 | 98% |
| Black or African-American | * | 0% | 7 | 0% |
| Other race or unspecified | 0 | 0% | 66 | 2% |
| Tobacco use indicated |  |  |  |  |
| Yes | 77 | 20% | 778 | 21% |
| Not indicated | 313 | 80% | 2932 | 79% |
| Alcohol use indicated |  |  |  |  |
| Yes | 23 | 6% | 236 | 6% |
| Not indicated | 367 | 94% | 3474 | 94% |
| Family History of SINT |  |  |  |  |
| First-degree relative (FDR) | 9 | 2% | 18 | <1% |
| Second-degree relative (SDR) | 12 | 3% | 31 | 1% |
| FDR and SDR | * | 1% | 0 | 0% |
| No FDR or SDR history of SINT | 365 | 94% | 3644 | 98% |
| Family history unknown for SDR | * | 1% | 17 | <1% |
| Vital Status |  |  |  |  |
| Living | 156 | 40% | 2159 | 58% |
| Deceased | 234 | 60% | 1551 | 42% |
| Mean age of death (std. dev.) | 75.40 (12.80) |  | 81.27 (9.94) |  |
| Median age of death (range) | 77 (24–101) |  | 83 (25–103) |  |
| Cancer Diagnosis (n = 390) |  |  |  |  |
| Mean age at diagnosis (std. dev.) | 62.98 (14.26) |  |  |  |
| Median age at diagnosis (range) | 64 (23–88) |  |  |  |
| Tumor Histology (n = 390) |  |  |  |  |
| 8240: Carcinoid tumor, malignant | 387 | 99% |  |  |
| 8241: enterochromaffin cell, carcinoid | * | 1% |  |  |
| ICDO: Tumor Location (n = 390) |  |  |  |  |
| 170: Duodenum | 37 | 9% |  |  |
| 171: Jejunum | 40 | 10% |  |  |
| 172: Ileum | 157 | 40% |  |  |
| 173: Meckel's diverticulum | 5 | 1% |  |  |
| 178: Overlapping lesions of small intestine | 7 | 2% |  |  |
| 179: Small intestine, not specified | 144 | 37% |  |  |
| Tumor grade (n = 113) |  |  |  |  |
| 1: Grade I—Well differentiated | 78 | 69% |  |  |
| 2: Grade II—Moderately differentiated | 31 | 27% |  |  |
| 3: Grade III—Poorly differentiated | * | 4% |  |  |
| Tumor stage (n = 381) |  |  |  |  |
| 0: In Situ | * | 0% |  |  |

(*Continued*)

**Table 2.** (Continued)

| | Cases (390) | | Controls (3710) | |
|---|---|---|---|---|
| 1: Localized | 137 | 36% | | |
| 2: Regional, direct extension only | 28 | 7% | | |
| 3: Regional, regional lymph nodes only | 53 | 14% | | |
| 4: Regional, direct extension and regional lymph nodes | 49 | 13% | | |
| 5: Regional, NOS | 33 | 9% | | |
| 7: Distant | 80 | 21% | | |

*Indicates a cell size of less than 5 cases.

was found for birth locales near orchards in Period 2, 1930 to 1944, a time when chemical pesticides were introduced (3.94 with 95% CI of 0.76 to 20.31, $p$ = 0.101). A significantly lower odds ratio was found for individuals born in a rural setting between 1945 and 1982 (OR = 0.58 with 95% CI = 0.36 to 0.96, $p$ = 0.033). Adjusting for family history of SINT, alcohol consumption and smoking did not significantly alter these results (S2 Table).

## Discussion

There is substantial interest and little information on the impact of early life or parental environmental exposures on adult health [15]. Early life exposures have been linked to the risk of subsequent development of prostate cancer and there is evidence that this may be related to epigenetic effects of environmental exposures [40]. Assessing early life impacts on rare cancers, such as SINT, is difficult as retrospective exposure data are needed for sufficient numbers of cases. Often, there is limited data on potential exposures during early life due to a lack of information on birth location and/or environmental conditions at those locations many decades into the past. Using a unique data resource, the UPDB, and historical information to reconstruct potential exposures we explored the link between early life exposures and the risk of developing SINT later in life. We found statistical evidence of clustering based on birth location, in locales coincident with areas where coal mining was the primary economic activity at the time of birth. Coal mining began in Utah in the 1850s and this region was historically dominated by coal mining and shipping with little agriculture or manufacturing [35]. Many mining towns were located at the mine site. These coal-company towns were slowly abandoned post-1942 as miners were no longer required to live in company housing and automobiles became more commonplace for commuting from nearby towns. From the 1950s to 1970s, many of the small coal mines in this area closed, as shown in Fig 1, and the population moved to more urban areas. As noted in the results, all of the cases in this spatial cluster were born before 1940. Additionally, the case-control analysis did find suggestive evidence of an elevated risk among individuals born in locales with coal, hard rock and uranium mining during Period 1 (1883–1929) when extraction practices would have led to high community exposures. Prior geospatial analyses have found residential proximity to coal mines in Illinois and Kentucky a risk factor for multiple cancers most notably colorectal and lung [41, 42]. These studies, however are not based on birth location, but residence at time of diagnosis. Although we speculate that coal mining was driving the observed cluster signal, other coincident factors could have been associated with elevated risk. A study from Canada found a higher incidence of neuroendocrine tumors in rural areas versus urban areas of residence [43]. The study's average age at diagnosis was 61 years which would translate to approximate birth years of 1936–1950. Our data on rural settlements suggests that the time period 1930–1944 had slightly elevated odds

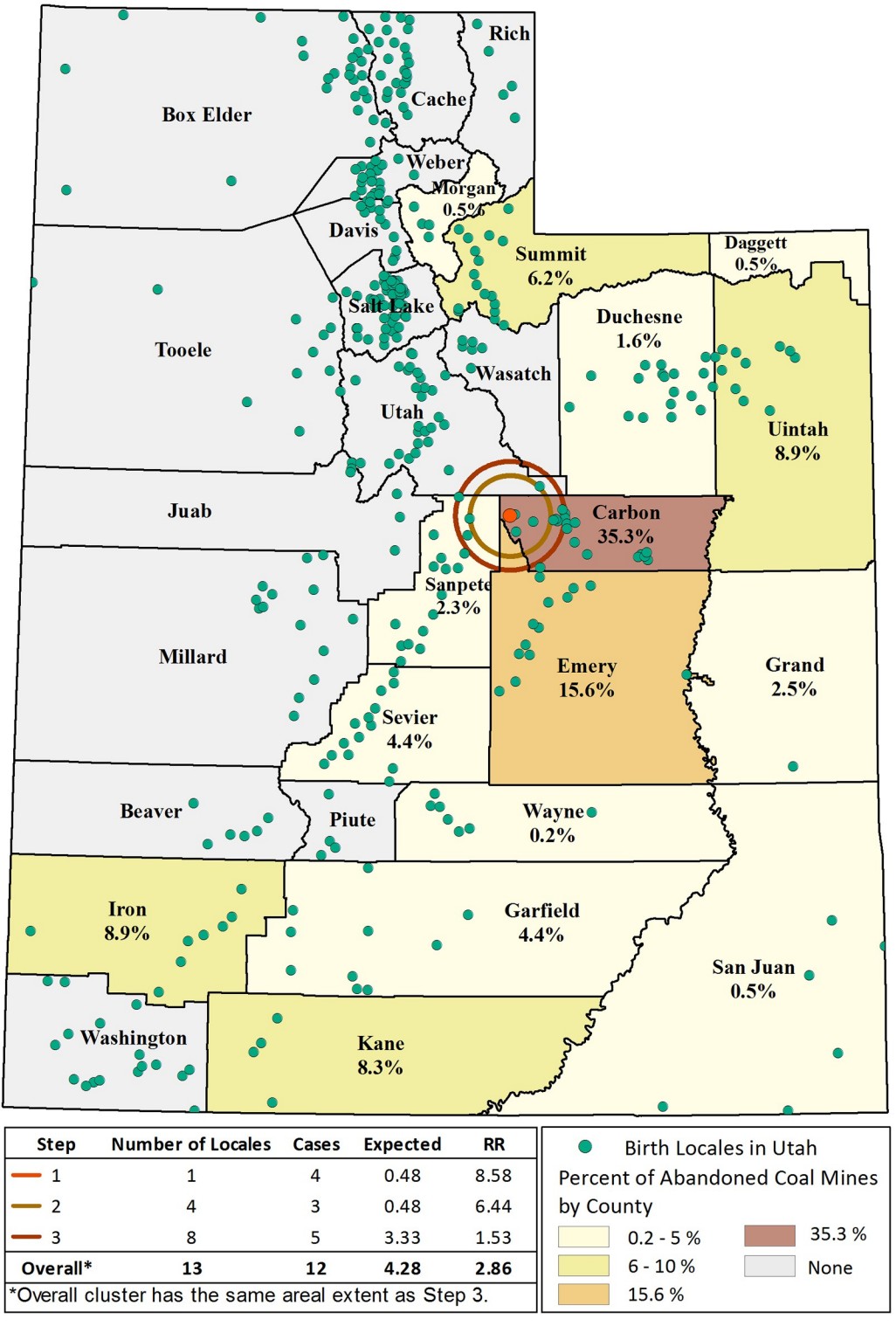

| Step | Number of Locales | Cases | Expected | RR |
|---|---|---|---|---|
| 1 | 1 | 4 | 0.48 | 8.58 |
| 2 | 4 | 3 | 0.48 | 6.44 |
| 3 | 8 | 5 | 3.33 | 1.53 |
| Overall* | 13 | 12 | 4.28 | 2.86 |
| *Overall cluster has the same areal extent as Step 3. | | | | |

● Birth Locales in Utah
Percent of Abandoned Coal Mines by County
- 0.2 - 5 %
- 6 - 10 %
- 15.6 %
- 35.3 %
- None

**Fig 1. Geographic analysis of SINT cases and controls.** The isotonic cluster signal and steps. The overall cluster has a 2.86 relative risk of SINT as shown in dark red. Overlaid on the Utah map are study defined birth locales for Utah (green circles) and the percent of currently abandoned coal mines by county.

**Table 3. Odds ratios for SINT from exposure to industries and settlements.**

| Locale Exposure | Period | % Cases Exposed | % Controls Exposed | UnadjustedOR | Lower CL | Upper CL | P-value |
|---|---|---|---|---|---|---|---|
| **MINING** | | | | | | | |
| Any Mining | Overall | 25.3 | 26.5 | 0.94 | 0.73 | 1.20 | 0.601 |
| | 1883–1929 | 41.1 | 38.0 | 1.16 | 0.81 | 1.65 | 0.414 |
| | 1930–1944 | 18.3 | 24.7 | 0.69 | 0.42 | 1.13 | 0.137 |
| | 1945–1982 | 13.4 | 15.3 | 0.86 | 0.50 | 1.47 | 0.582 |
| Hard Rock | Overall | 25.3 | 23.8 | 1.09 | 0.85 | 1.39 | 0.507 |
| | 1883–1929 | 40.4 | 33.3 | 1.38 | 0.97 | 1.97 | 0.075 |
| | 1930–1944 | 18.3 | 21.6 | 0.82 | 0.50 | 1.34 | 0.432 |
| | 1945–1982 | 14.2 | 15.2 | 0.93 | 0.55 | 1.56 | 0.781 |
| Coal | Overall | 4.1 | 3.8 | 1.08 | 0.64 | 1.83 | 0.763 |
| | 1883–1929 | 6.8 | 3.5 | 1.95 | 0.97 | 3.92 | 0.061 |
| | 1930–1944 | 2.6 | 4.9 | 0.52 | 0.16 | 1.68 | 0.275 |
| | 1945–1982 | 2.4 | 3.0 | 0.78 | 0.24 | 2.57 | 0.684 |
| Uranium | Overall | 1.8 | 1.0 | 1.75 | 0.77 | 3.99 | 0.181 |
| | 1883–1929 | 2.7 | 1.0 | 2.70 | 0.86 | 8.48 | 0.088 |
| | 1930–1944 | 0.9 | 1.2 | 0.73 | 0.09 | 5.69 | 0.761 |
| | 1945–1982 | 1.6 | 0.9 | 1.75 | 0.38 | 8.06 | 0.473 |
| Minerals | Overall | 3.1 | 4.0 | 0.77 | 0.43 | 1.41 | 0.403 |
| | 1883–1929 | 3.4 | 5.2 | 0.65 | 0.26 | 1.64 | 0.361 |
| | 1930–1944 | 4.3 | 4.4 | 0.98 | 0.38 | 2.49 | 0.960 |
| | 1945–1982 | 1.6 | 2.1 | 0.74 | 0.17 | 3.15 | 0.685 |
| **INDUSTRY** | | | | | | | |
| Heavy Industry | Overall | 14.4 | 15.9 | 0.88 | 0.65 | 1.19 | 0.409 |
| | 1883–1929 | 9.6 | 12.1 | 0.71 | 0.39 | 1.29 | 0.266 |
| | 1930–1944 | 20.9 | 17.7 | 1.23 | 0.76 | 1.97 | 0.403 |
| | 1945–1982 | 14.2 | 18.4 | 0.73 | 0.44 | 1.24 | 0.247 |
| Smelting | Overall | 2.8 | 3.1 | 0.93 | 0.50 | 1.74 | 0.821 |
| | 1883–1929 | 2.7 | 3.5 | 0.80 | 0.29 | 2.24 | 0.673 |
| | 1930–1944 | 4.3 | 3.6 | 1.20 | 0.46 | 3.12 | 0.703 |
| | 1945–1982 | 1.6 | 2.1 | 0.75 | 0.18 | 3.21 | 0.701 |
| Construction | Overall | 5.9 | 4.8 | 1.24 | 0.79 | 1.95 | 0.358 |
| | 1883–1929 | 10.3 | 5.5 | 1.98 | 1.09 | 3.60 | 0.024 |
| | 1930–1944 | 4.3 | 4.4 | 0.98 | 0.38 | 2.53 | 0.969 |
| | 1945–1982 | 2.4 | 4.5 | 0.51 | 0.16 | 1.68 | 0.269 |
| **AGRICULTURE** | | | | | | | |
| Crop | Overall | 3.9 | 4.4 | 0.85 | 0.50 | 1.47 | 0.568 |
| | 1883–1929 | 6.2 | 6.1 | 0.98 | 0.48 | 1.99 | 0.947 |
| | 1930–1944 | 4.3 | 4.6 | 0.95 | 0.37 | 2.42 | 0.916 |
| | 1945–1982 | 0.8 | 2.4 | 0.33 | 0.05 | 2.46 | 0.282 |
| Orchard | Overall | 0.5 | 0.8 | 0.66 | 0.16 | 2.76 | 0.566 |
| | 1883–1929 | 0.0 | 1.0 | 0.00 | 0.00 | | 0.982 |
| | 1930–1944 | 1.7 | 0.4 | 3.94 | 0.76 | 20.31 | 0.101 |
| | 1945–1982 | 0.0 | 0.8 | 0.00 | 0.00 | | 0.985 |
| Livestock | Overall | 9.3 | 9.3 | 0.99 | 0.69 | 1.43 | 0.971 |
| | 1883–1929 | 12.3 | 13.1 | 0.92 | 0.55 | 1.55 | 0.755 |
| | 1930–1944 | 12.2 | 9.0 | 1.42 | 0.78 | 2.56 | 0.250 |
| | 1945–1982 | 3.1 | 5.3 | 0.59 | 0.21 | 1.64 | 0.308 |

*(Continued)*

**Table 3.** (Continued)

| Locale Exposure | Period | % Cases Exposed | % Controls Exposed | UnadjustedOR | Lower CL | Upper CL | P-value |
|---|---|---|---|---|---|---|---|
| **TYPE OF SETTLEMENT** | | | | | | | |
| City | Overall | 40.5 | 42.2 | 0.94 | 0.75 | 1.16 | 0.552 |
| | 1883–1929 | 36.3 | 35.0 | 1.09 | 0.76 | 1.57 | 0.621 |
| | 1930–1944 | 29.6 | 36.1 | 0.75 | 0.49 | 1.13 | 0.170 |
| | 1945–1982 | 55.1 | 55.9 | 0.95 | 0.66 | 1.38 | 0.806 |
| Rural | Overall | 36.9 | 38.5 | 0.91 | 0.73 | 1.14 | 0.407 |
| | 1883–1929 | 48.6 | 48.3 | 0.98 | 0.69 | 1.39 | 0.918 |
| | 1930–1944 | 45.2 | 42.2 | 1.13 | 0.77 | 1.66 | 0.543 |
| | 1945–1982 | 15.7 | 24.4 | 0.58 | 0.36 | 0.96 | 0.033 |
| Commerce | Overall | 37.9 | 37.6 | 1.03 | 0.82 | 1.29 | 0.824 |
| | 1883–1929 | 17.8 | 24.3 | 0.67 | 0.43 | 1.06 | 0.091 |
| | 1930–1944 | 43.5 | 38.4 | 1.24 | 0.84 | 1.83 | 0.278 |
| | 1945–1982 | 55.9 | 51.6 | 1.17 | 0.81 | 1.70 | 0.402 |
| Connected | Overall | 31.7 | 30.7 | 1.08 | 0.85 | 1.37 | 0.530 |
| | 1883–1929 | 13.0 | 18.7 | 0.65 | 0.39 | 1.09 | 0.105 |
| | 1930–1944 | 34.8 | 30.5 | 1.23 | 0.82 | 1.84 | 0.326 |
| | 1945–1982 | 50.4 | 44.3 | 1.29 | 0.90 | 1.87 | 0.166 |
| Shipping | Overall | 10.3 | 11.3 | 0.91 | 0.64 | 1.28 | 0.585 |
| | 1883–1929 | 7.5 | 11.6 | 0.63 | 0.34 | 1.20 | 0.161 |
| | 1930–1944 | 14.8 | 13.0 | 1.17 | 0.68 | 2.01 | 0.575 |
| | 1945–1982 | 9.4 | 9.4 | 1.00 | 0.52 | 1.90 | 0.992 |

ratio (1.13) but this was not significant. An interesting parallel is our observation of a 3.9 increased odds of SINT in the orchard industry for this same second period which might point to the advent of chemical pesticides.

Limitations of this study were our use of a single residential location to characterize early life exposures, determination of historical community exposures, multiple hypothesis testing issues, and small sample size. Place of birth has benefits in that it is often the residence throughout childhood, so it is a reasonable snapshot of early life exposures. While ascertaining potential exposures in the early part of the twentieth century is challenging, several factors suggest that exposures may have been more homogenous and consistent over time. For example, consistent residential mobility was generally less than the present day, settlements had only one or two primary economic activities (e.g., mining or manufacturing), and these activities did not change rapidly over short periods of time. Future studies may be able to use complete residential histories made possible by the UPDB to characterize exposures over the life course as only a small proportion of all available addresses (11%) are attributable to birth certificates. We were also concerned that relatedness of cases, a known risk factor, could overlap with geographic clustering. When analysis of exposure was adjusted for family history of SINT there were no significant changes, suggesting that the risk exposures we identified were independent of familial risk. Although exposure analysis was adjusted for indication of tobacco use, data on tobacco use prior to 1996 was limited to contributing cause of death information recorded on a Utah death certificate. Thus, it is likely that for some subjects, indication of tobacco use was misclassified, particularly for case and control subjects who died before 1996 in which exposure data were limited.

Our study applied a novel set of methods to use in-depth historical research to characterize the early life environment of cases and controls. Because birth certificate records are a

comprehensive resource across time, very few of these rare cancers were excluded due to a lack of birth locale. Future studies that capture the cumulative lifetime exposures for an individual, while complex, could provide valuable insights but would be particularly challenging for studies of rarer cancers that may lack sufficient numbers. Unlike birth certificates, other sources of residential history are comparatively incomplete and are limited in scope in terms of years covered, thus many cases would likely be excluded. The methods used here, along with cumulative exposures, could practically be applied to the more common cancers or in large populations of rarer cancers, where numbers would be sufficient for study.

## Conclusion

Using cases and birth cohort controls, we report geographic clustering of a birth locale associated with greater than 2-fold elevated risk of SINT, located central to historic coal mining communities. A retrospective analysis of industry and mining exposures further suggest birth/early life residence near construction industry, uranium mining and coal mining confers an elevated risk, specifically during the earliest time period when there were no regulations to reduce exposures to workers or the public.

## Supporting information

**S1 Table. Sources of data used to characterize environmental and economic conditions of birth locales.**
(DOCX)

**S2 Table. Unadjusted and adjusted odds ratios for SINT from exposure to industries and settlements.**
(XLSX)

## Acknowledgments

We thank the University of Utah Center for Clinical and Translational Science (funded by NIH Clinical and Translational Science Awards), the Pedigree and Population Resource, University of Utah Information Technology Services and Biomedical Informatics Core for establishing the Master Subject Index between the Utah Population Database, the University of Utah Health Sciences Center and Intermountain Health Care.

## Author Contributions

**Conceptualization:** James VanDerslice, Karen Curtin, Lisa A. Cannon-Albright, Deborah W. Neklason.

**Data curation:** James VanDerslice, Marissa C. Taddie, Karen Curtin, Caroline Miller, Zhe Yu, Rachael Hemmert.

**Formal analysis:** James VanDerslice, Marissa C. Taddie, Karen Curtin, Caroline Miller, Lisa A. Cannon-Albright, Deborah W. Neklason.

**Funding acquisition:** Deborah W. Neklason.

**Investigation:** James VanDerslice, Karen Curtin, Caroline Miller, Zhe Yu, Rachael Hemmert, Lisa A. Cannon-Albright, Deborah W. Neklason.

**Methodology:** James VanDerslice, Rachael Hemmert, Lisa A. Cannon-Albright, Deborah W. Neklason.

**Project administration:** Lisa A. Cannon-Albright, Deborah W. Neklason.

**Resources:** Zhe Yu, Rachael Hemmert, Lisa A. Cannon-Albright, Deborah W. Neklason.

**Software:** James VanDerslice.

**Supervision:** James VanDerslice, Deborah W. Neklason.

**Visualization:** Marissa C. Taddie.

**Writing – original draft:** James VanDerslice, Karen Curtin, Lisa A. Cannon-Albright, Deborah W. Neklason.

**Writing – review & editing:** James VanDerslice, Marissa C. Taddie, Karen Curtin, Caroline Miller, Zhe Yu, Rachael Hemmert, Lisa A. Cannon-Albright, Deborah W. Neklason.

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
