## [Decision Letter · Decision Letter 0]

7 Feb 2020

PONE-D-20-00235

Early life exposures associated with risk of small intestinal neuroendocrine tumors

PLOS ONE

Dear Dr. Neklason,

Thank you for submitting your manuscript to PLOS ONE. After careful consideration, we feel that it has merit but does not fully meet PLOS ONE’s publication criteria as it currently stands. Therefore, we invite you to submit a revised version of the manuscript that addresses the points raised during the review process.

We would appreciate receiving your revised manuscript by Mar 23 2020 11:59PM. To enhance the reproducibility of your results, we recommend that if applicable you deposit your laboratory protocols in protocols.io, where a protocol can be assigned its own identifier (DOI) such that it can be cited independently in the future. For instructions see: http://journals.plos.org/plosone/s/submission-guidelines#loc-laboratory-protocols

We look forward to receiving your revised manuscript.

Kind regards,

Jaymie Meliker, Ph.D.

Academic Editor

PLOS ONE

Journal Requirements:

2. To comply with PLOS ONE submission guidelines, in your Methods section, please provide additional information regarding your statistical analyses. For more information on PLOS ONE's expectations for statistical reporting, please see https://journals.plos.org/plosone/s/submission-guidelines.#loc-statistical-reporting.

Reviewers' comments:

Reviewer's Responses to Questions

**Comments to the Author**

1. Is the manuscript technically sound, and do the data support the conclusions?

Reviewer #1: Yes

Reviewer #2: Yes

2. Has the statistical analysis been performed appropriately and rigorously? 

Reviewer #1: Yes

Reviewer #2: Yes

3. Have the authors made all data underlying the findings in their manuscript fully available?

Reviewer #1: Yes

Reviewer #2: No

4. Is the manuscript presented in an intelligible fashion and written in standard English?

Reviewer #1: Yes

Reviewer #2: Yes

5. Review Comments to the Author

Reviewer #1: I appreciated the opportunity to review this work. While SINT is not as high a priority as other cancers, I applaud the exercise of spatial cluster analysis coupled with historical data to explore how pre-/ante-natal exposures may impact cancer susceptibility. Data regarding environmental toxins (endocrine-disrupting pesticides, industrial waste, etc.) and their influence are hard to come by and such methodologies need further exercise and refinement. Still, there are multiple aspects of work (relatively minor) which need to be addressed.

1. I would remove the non-significant OR from the abstract and more simply indicate that other exposures approach significance. While the 0.05 is an arbitrary level, it is the standard convention.

2. Line 50 - Perhaps rearrange to say "...epidemiological studies have been limited." (trivial I know)

3. Line 57 - Indicate the countries included in the meta-analysis. The rest of the manuscript is based on Utah exposure, which might be different than other places.

4. Lines 60-61 - Re-word the sentence so its clear that a large proportion of residents hold religious proscriptions. As written it almost comes across as promulgated by the state of Utah.

5. Line 72 - While you define eligible life events later, please provide a few examples here ("...with a life event (e.g. birth, marriage, etc.)...").

6. Line 89 - What proportion of the Utah population is part of the >11 people?

7. Line 96 - Does "..this year." mean 2019?

8. Line 105 - Seems close to half of cases are excluded. What types of bias might be represented here (e.g. lower income more likely to remain in state)?

9. Lines 110-113 - Couple things

9a - Make clear you have alcohol/tobacco data for each/all cases and controls.

9b - How are/were multiple records resolved (e.g. an individual might have 10 entries for alcohol use over 20 years, ranging from 0 consumption to 3 drinks/day)?

9c - What type(s) of bias and possible error might be associated with lack of data prior to 1996? Some people have died before this time.

10. Line 164 - How were social conditions and economic activities for each locale determined and consolidated? Assessed for each year, averaged over a time period, other? Seems a factory might have a large economic splash for 10 years in a period, but not the rest before opening/after closing.

11. Table 1 - better define how rural/urban were determined.

12. Lines 179-181 - The justification/definitions for the 3 time periods could be strengthened, and maybe mover earlier in Methods. Up to this point it almost seems like year by year modeling.

13. Line 201 - Indicate "...and three ADDITIONAL cases."

Reviewer #2: This is a well written manuscript that presents original research investigating potential environmental risk factors for small intestinal neuroendocrine tumors (SINT). SINT are rare but have been increasing in incidence over the past 40 years and little is known about the etiology. The authors conducted a record linkage case-control study using the Utah Population Database (UTPD), which contains residential histories and life event data. Using a spatial scan statistic, they identified a cluster based on birth residence in an area that historically had coal mines. Using residence location at birth they identified increased risk for SINT among those living near construction and mining operations before 1940. Lower risk was observed in rural areas among those born in a later period (1945-1985). The authors went to considerable effort to link birth residences in their study to historical locations of mines, construction sites, and agriculture and this appears to be the first effort to link these historical data to cancer incidence data in Utah. This exploratory analysis of SINT incidence in relation to characteristics of the birth residence makes an important contribution to the limited literature on environmental exposures and these cancers. My main comments are about providing more detail on the both the historical environmental database and the UTPD, which are important for understanding and interpreting their results.

Major comments

1. I would like to see more details about the exposure database, which was reconstructed over an approximately 100-year period. Please describe how accurately you were able to locate the industries and agricultural areas and the completeness of your data sources over time and by industry type. What were your sources of the agricultural data back through time? Were USGS land cover maps used for the most recent period? A supplemental table documenting some of these details would be helpful for understanding the data sources and potential gaps or uncertainties in these spatial data.

2. How complete are the residential history data in the UTPD? Please describe this for the study population and briefly describe factors associated with completeness (e.g., gender, birth year, etc). Although the authors reference several publications that use the UTPD, I think it’s appropriate to include more details about the database including the relative availability of residential data sources back through time. Please clarify if a LexisNexis search was used to obtain some of the historical residence information and if not, why wasn’t this done?

3. Little is known about risk factors for SINT, particularly whether early life is a critical time period for exposures or if long-term exposures are most important. The authors mention that the use of one residence location to characterize exposure is a limitation of their study. They also point out the importance of evaluating cumulative exposures in future studies, with which I agree. No doubt this would be challenging to do for SINT since it may substantially increase the geographic extent of their exposure assessment. However, given that at least some residential history data are available for their cases and controls, I would like to see information on the percentage of the residence histories represented by the birth residence. Also, a sensitivity analysis limited to those with longer duration at this address would be informative.

4. Related to comment #3, the authors mention in the discussion that an evaluation of cumulative exposures would be particularly challenging for rarer cancers due to power issues. Is this because many cases would be excluded due to a lack of substantial residence histories? Please clarify.

5. The introduction mentions increasing incidence of colorectal cancer in younger adults and the hypothesis that this increase is related to early life exposures. Did you conduct analyses separately for younger and older cases of SINT? It would be interesting to know if risk differed by age at diagnosis. Further, were there any differences by gender? Higher risk among men would suggest that occupational exposures might be important.

6. Cases were from 1948 through 2014. However, cancer data is incomplete before 1966 when the tumor registry was established. How confident are you that the cases prior to 1966 are representative of all SINT cases in this earlier period? Can you conduct sensitivity analyses excluding cases and their matched controls prior to 1966?

Specific Comments:

Abstract

1. include the years of case ascertainment and the age range of the cases. Conclusion sentence seems too strong given the exploratory nature of this analysis.

Introduction

2. Tobacco use is listed as one of the factors that may contribute to the rise in SINT; however, tobacco use has gone down over time. Please clarify this point.

Methods

3. Page 5. Case and control selection. Almost ½ of the 745 cases identified in the Utah Cancer Registry did not meet the inclusion criteria. Please clarify the reasons for exclusion. Provide information on the percent of cases and controls for whom hospital location was used as a proxy for birth residence.

4. Using the UTM data that is available on some cases and controls, can you describe the positional errors in the locale centroids? Does this differ for urban and rural areas? Please comment on how these errors might result in exposure misclassification.

5. Since each birth locale can have multiple industries during the same time period (page 8), is it feasible to compute a metric that accounts for multiple industries? How common was a multiple industry exposure scenario?

Discussion

6. The large amount of missing data for smoking and alcohol intake should be acknowledged as a limitation. How was 'not indicated' information handled in the adjusted analysis? Are the percentages of smoking and alcohol intake in controls comparable to the Utah population in this time period/age range?

7. Did the authors consider how radioactive fallout from nuclear tests in the 1940s to 1960s might have exposed their study population?

6. PLOS authors have the option to publish the peer review history of their article (what does this mean?). If published, this will include your full peer review and any attached files.

Reviewer #1: No

Reviewer #2: No

---

## [Author Response · Author response to Decision Letter 0]

30 Mar 2020

Dear Editors and reviewers, 

Thank you for the insightful and helpful reviews and the opportunity to resubmit a revised version of the manuscript. 

Responses to editor and reviewer comments are included in the cover letter.

---

## [Editor Report · Decision Letter 1]

6 Apr 2020

Early life exposures associated with risk of small intestinal neuroendocrine tumors

PONE-D-20-00235R1

Dear Dr. Neklason,

We are pleased to inform you that your manuscript has been judged scientifically suitable for publication and will be formally accepted for publication once it complies with all outstanding technical requirements.

With kind regards,

Jaymie Meliker, Ph.D.

Academic Editor

PLOS ONE
---

## [Editor Report · Acceptance letter]

10 Apr 2020

PONE-D-20-00235R1 

Early life exposures associated with risk of small intestinal neuroendocrine tumors 

Dear Dr. Neklason:

I am pleased to inform you that your manuscript has been deemed suitable for publication in PLOS ONE. Congratulations! Your manuscript is now with our production department. 

With kind regards,

on behalf of

Dr. Jaymie Meliker 

Academic Editor

PLOS ONE